# Metabolic Function and Therapeutic Potential of CD147 for Hematological Malignancies: An Overview

**DOI:** 10.3390/ijms25179178

**Published:** 2024-08-23

**Authors:** Isabella Spinello, Catherine Labbaye, Ernestina Saulle

**Affiliations:** Istituto Superiore di Sanità, National Center for Drug Research and Evaluation, 00161 Rome, Italy; isabella.spinello@iss.it (I.S.); catherine.labbaye@iss.it (C.L.)

**Keywords:** hematological malignancies, metabolism, CD147

## Abstract

Hematological malignancies refer to a heterogeneous group of neoplastic conditions of lymphoid and hematopoietic tissues classified in leukemias, Hodgkin and non-Hodgkin lymphomas and multiple myeloma, according to their presumed cell of origin, genetic abnormalities, and clinical features. Metabolic adaptation and immune escape, which influence various cellular functions, including the proliferation and survival of hematological malignant tumor cells, are major aspects of these malignancies that lead to therapeutic drug resistance. Targeting specific metabolic pathways is emerging as a novel therapeutic strategy in hematopoietic neoplasms, particularly in acute myeloid leukemia and multiple myeloma. In this context, CD147, also known as extracellular matrix metalloproteinase inducer (EMMPRIN) or Basigin, is one target candidate involved in reprograming metabolism in different cancer cells, including hematological malignant tumor cells. CD147 overexpression significantly contributes to the metabolic transformation of these cancer cells, by mediating signaling pathway, growth, metastasis and metabolic reprogramming, through its interaction, direct or not, with various membrane proteins related to metabolic regulation, including monocarboxylate transporters, integrins, P-glycoprotein, and glucose transporter 1. This review explores the metabolic functions of CD147 and its impact on the tumor microenvironment, influencing the progression and neoplastic transformation of leukemias, myeloma, and lymphomas. Furthermore, we highlight new opportunities for the development of targeted therapies against CD147, potentially improving the treatment of hematologic malignancies.

## 1. Introduction

Hematologic malignancies (HMs) represent a wide range of malignant tumors of the blood and lymphatic system that share pathological alterations capable of inducing radical transformations in cells, characterized by genetic, histological, and metabolic alterations associated with uncontrolled cell proliferation and high resistance to cell death [1,2,3,4,5]. From three main groups including leukemias, lymphomas, and multiple myeloma, HM can be categorized into different subgroups, such as acute leukemias, chronic leukemias, myelodysplastic syndromes (MDSs), myeloproliferative neoplasms (MPNs), non-Hodgkin lymphomas, and classic Hodgkin lymphomas [2,3,4,5]. Aberrant metabolism can significantly affect the immune microenvironment and thus modulate the immune cellular response of hematological malignant tumor cells, contributing to disease progression [6]. 

Increasing progress has been made for the treatment of HMs, involving drugs classified as chemotherapy agents, whose activity is due to the disruption of the mitotic and/or DNA replication pathways and targeted agents, which inhibit the molecular targets involved in cancer cell growth and spread. New targeted therapies with small molecule inhibitors, but also monoclonal antibodies, bispecific T-cell engagers, antibody–drug conjugates, recombinant immunotoxins, and, more recently, chimeric antigen receptor T (CAR-T) cells have improved the clinical outcomes for blood cancers, especially for older patients and/or patients with relapsed or refractory hematological malignancies who do not respond to standard treatments [7,8,9]. 

Clinical and preclinical studies have shown that targeting metabolic features of hematologic malignancies is an appealing therapeutic approach [6,7,8]. As tumor cells utilize glucose aerobically as an energy source and an intermediate for other metabolic pathways, the most common metabolic change found in HMs is the increased glucose consumption of the tumor cells through aerobic glycolysis (“Warburg effect”), which leads to the increased production and accumulation of lactate [9]. Lactate, derived from glucose fermentation, is excreted from the cell. The inhibition of lactate production or transport of lactate out of the cell are considered two potential strategies to directly target the Warburg effect in hematological malignancies [10,11].

CD147, an extracellular matrix metalloproteinase inducer (EMMPRIN) also called Basigin [12,13], is a multifunctional glycoprotein whose overexpression in various cancers is associated with poor prognosis, including in HMs [13,14,15,16,17]. CD147 that interacts with membrane proteins related to metabolic regulation, including monocarboxylate transporters (MCT1 and MCT4), integrins such as CD98, P-glycoprotein (P-gp), and glucose transporter 1 (GLUT1), significantly contributes to the metabolic transformation of neoplastic cells [14,15,16,17]. CD147 is involved in the regulation of the main metabolic pathways, such as glycolysis, oxidative phosphorylation, and lipolysis [13,14,18]. Furthermore, the immunomodulatory function of CD147 has emerged to be particularly relevant in cancer [19]. Then, CD147’s overexpression can be used as a biomarker in HMs [14,15,16,17,18,20], and its protein partners represent a potential therapeutic target in the treatment of hematologic malignancies for the development of new therapies aimed at identifying and manipulating specific metabolic vulnerabilities of the different main groups, leukemia, lymphoma, and multiple myeloma, of HMs that support neoplastic cells proliferation. 

In this review, we will examine recent advances in the characterization of metabolic pathways influenced by CD147 and its partners, and in the development of new therapies for hematological malignancies. The review will also highlight the role of CD147 in tumor glucose metabolism reprograming and its interplay with the tumor microenvironment (TME). 

## 2. CD147 

### 2.1. CD147 Structure and Expression

CD147 is a transmembrane glycoprotein that belongs to the superfamily of immunoglobulins and is present in four isoforms (CD147/Basigin-1, -2, -3, and -4). Among these, isoform BSG-2, with two immunoglobulin domains, is the most abundant and most studied form, ubiquitously expressed in humans by many types of cells, including hematopoietic, epithelial, and endothelial cells [13,14]. 

CD147 is a potent inducer of extracellular matrix metalloproteinase (EMMPRIN) that plays a key role in the regulation of extracellular matrix (ECM) remodeling during physiological and pathological processes, such as wound healing, fibrotic and inflammatory-related diseases [13,21,22], and cancer, including hematological neoplasia, such as AML and multiple myeloma, in myelodysplastic syndrome with a 5q deletion, where CD147 expression levels have a prognostic value [13,14,16,17,18,19,20]. But, CD147 is also a multifunctional protein that exerts pleiotropic functions by interacting with various binding partners, such as monocarboxylate transporters, MCT1-4, involved in metabolic pathways and T-lymphocyte activation; cyclophilin A involved in adhesion, viral invasion, and inflammation; E-selectin; integrins; and caveolin-1 involved in adhesion [13,14]. CD147, expressed on the cell surface of leucocytes, plays a crucial role in the recruitment of innate or adaptative immune cells required to fight virus infections and consequent inflammation [14,23,24]. 

The glycoprotein CD147 has a molecular weight between 28 and 60 KDa, depending on the level of glycosylation at the N-terminal end, which consists of three N-bound glycosylation sites at Asn (Asn44, Asn 152, and Asn186). CD147 N-glycans determine the different molecular weights, low LG-CD147 and a high level of N-glycans, in HG-CD147 form. The level of glycosylation achieved by CD147 and the ratio between the two glycosylated forms, LG-CD147 and HG-CD147, are two characteristics that play a fundamental role in affecting its biological functions; the N-glycosylation of CD147 is necessary for stabilizing the CD147 glycoprotein, transporting CD147 to the cell membrane, modulating the secretion of MMPs, and finally initiating extracellular matrix remodeling. Furthermore, the different N-glycosylation on CD147 influences its association with protein partners [25,26,27]. N-glycosylation on CD147 is essential for its association with E-selectin during leukocyte infiltration in renal inflammation [28] and during the inflammatory response observed in CD147 overexpression-associated diseases [28,29]. Conversely, glycosylation can also inhibit the interaction between proteins, as Tang et al. have shown, by the increased interaction of the deglycosylated form of CD147 with caveolin-1, thus indicating that glycosylation on CD147 interferes with its interaction with caveolin-1 [29,30]. The possible role of the N-glycosylation of CD147 in its interaction with other proteins, such as MCTs, integrins, and cyclophilins, has yet to be investigated with further studies. Importantly, the association of the altered glycosylation of CD147 and multidrug resistance in human leukemia [31], the profound influence of the ratio of the two glycosylated forms, HG-CD147 and LG-CD147, of CD147 on neoplastic diseases, and the correlation of the HG-CD147 form found in acute lymphoblastic leukemia (ALL) and its recurrence [32] have been demonstrated. Then, CD147 glycosylation, which may be used as a marker to predict the clinical prognosis of cancer, represents a potential therapeutic target in cancers overexpressing the HG-CD147 form, and the development of new target-specific inhibitors for CD147 glycosylation could significantly improve the prognosis of cancer patients [27] (Table 1).

CD147 is expressed in both membrane-bound and soluble forms [13,14,33]. The soluble form is considered to act in a paracrine fashion by binding to the membrane-bound form (i.e., dimerization), while CD147 dimerization is essential for promoting tumor invasion through its involvement in the production/activation of matrix metalloproteinases (MMPs) [34]. The soluble form of CD147 (sCD147) can be secreted from cells and released into the microenvironment, also as a full-length protein released via microvesicles [35]. sCD147 detection in the serum or plasma obtained from several types of tumors, associated with tumor growth, metastasis formation, and chemoresistance, confers to CD147 its role as a clinical biomarker in several cancers, including hematological malignancies [36,37,38,39,40]. Furthermore, sCD147 influences tumor cell proliferation through its interaction with the membrane-bound form of CD147. The internalization of sCD147 has also been shown to stimulate sCD147 production via positive feedback [36,37,38]. The peculiar structural characteristics of CD147 influence its ability to interact with different extracellular, intracellular, and membrane proteins, ensuring its involvement in multiple biological process, including cellular transport, adhesion, cell proliferation, angiogenesis, metastasis, and apoptosis [13,14,15,16]. Although there are several/numerous studies on the role played by CD147 in various diseases and solid tumors [16,18], only recently has in-depth research clarified the biological role of this glycoprotein in normal hematopoiesis and in the development of hematological malignancies [13,14,15,24].

In a first study, the expression of CD147 has been shown to be induced in activated leukocytes, such as granulocytes, lymphocytes, and macrophages [41]. In our recent work, we find that CD147 is expressed in normal CD34+ hematopoietic progenitor cells (HPCs) and downregulated during the monocytic and granulocyte differentiation of HPCs in “in vitro” cultures [15]. In blasts related to different subtypes of AML, CD147 is overexpressed compared to its normal counterpart. Specific inhibition of its activity with AC-73 inhibits proliferation and induces autophagy, demonstrating a critical role in the proliferation of leukemia cells [15]. In a subsequent study, to further characterize the downstream CD147 signaling pathway/network, we also extended our analysis on the expression and function of two partner molecules, MCT1 and MCT4, in hematopoietic progenitor cells, in different leukemia cell lines, and in primary leukemic blasts from AML patients [15,42]. We found an inverse correlation between MCT1 and MCT4 expression levels in leukemia cells and showed that MCT4 overexpression is associated with a poor prognosis in AML patients [42]. Interestingly, CD147 is expressed at the level of CD34/CD371 AML cells, previously described for their leukemia-initiating (LIC) properties [15]. Another study demonstrates that CD147 knockdown in AML cells suppressed growth and proliferation, reduced protein levels of NF-κB (p65) and Bcl-2, Bcl-xL, and lowered the anti- and pro-apoptotic protein ratios of Bcl-2/Bax and Bcl-xL/Bax, clarifying the role of CD147 in the signaling pathways involved in the development of leukemia [43]. Schmidt’s study reports an overexpression of CD147 also in several lymphoma and leukemia cell lines, as well as in normal activated T cells [44]. Protein analysis showed an increase in the high glycosylation form (HG-CD147) in ALK+ anaplastic large-cell lymphoma (ALCL) cell lines, ALK− cell lines, and several B-cell lymphoma cell lines. The weakest expression of the HG-CD147 form has been detected in mantle cell lymphoma and T-cell acute lymphoblastic lymphoma cell lines. The low glycosylated form (LG-CD147) was strongly expressed in Mac-1 and HeLa, weakly expressed in several cell lines of human diffuse large B-cell lymphomas, Raji (Burkitt lymphoma) cells, and Granta B cell leukemia. Interestingly, in resting CD3+ T-cell samples, the authors did not detect any expression of CD147 at the protein level; this result was confirmed by the immunohistochemical analysis of normal T cells in the interfollicular area in normal lymph nodes, where CD147 expression was negative in contrast to the positive expression in the B cells of the germinal center and mantle areas [44]. Furthermore, CD147 is overexpressed in myelodysplastic syndrome (MDS) erythroid cells with 5q deletion and in multiple myeloma (MM), where CD147 expression levels have a prognostic value and are necessary for the proliferation of MM cells [17,19,45]. 

**Table 1 ijms-25-09178-t001:** CD147 expression data in different hematological malignancies.

CD147 Glycosylated Form	Hematological Malignancies	Role	Ref.
**HG-CD147**	CML	Multidrug resistance	[31]
**HG-CD147/LG-CD147** **Ratio**	ALL	Associated with relapse ALL	[32]
**HG-CD147**	AML	Proliferation and multidrug resistance	[15]
**HG-CD147**	AML	Proliferation	[43]
**HG-CD147**	ALK+ ALCL ALK− ALCL B-cell lymphoma	Proliferation	[44]
**Weak HG-CD147**	Mantle cell lymphoma T-cell lymphoma	Proliferation	[44]
**HG-CD147**	Multiple myeloma, CLL, and lymphoplasmacytic lymphoma (LPL)	Proliferation	[17]
**HG-CD147**	Multiple myeloma del(5q) MDS	Proliferation	[19]
**HG-CD147**	Multiple myeloma	Survival	[45]

### 2.2. CD147 Functions: Its Interaction with MCTs and Other Partners

The CD147 glycoprotein was initially known as an matrix metalloproteinase (MMP) inducer, capable of stimulating fibroblasts and endothelial cells to facilitate tumor invasion, metastasis, and angiogenesis [46,47]. Several studies have highlighted that CD147 also plays a role in several other functions and can interact with different partner proteins to regulate multiple signaling pathways [13,14] (Figure 1). Furthermore, CD147 is involved in angiogenesis through the stimulation of vascular endothelial growth factor (VEGF) production [20,47]. Connected to cellular metabolic regulation, one of the prevalent biological functions carried out by CD147 concerns the regulation of the expression and the activity of monocarboxylate transporters-1 (MCT-1) and -4 (MCT-4) to form complexes on the membrane to transport lactic acid produced from glycolysis [48]. This function is particularly critical in the context of neoplastic development, as CD147 supports glycolysis in the presence of oxygen in tumor cells [49]. MCTs catalyze the transport of lactate, pyruvate, and ketone bodies across the plasma membrane. There are four isoforms (MCT1–MCT4) with different modes of expression and distinct substrate affinities [50]. Mechanistically, CD147 serves as a chaperone required for the plasma membrane translocation of MCTs [51]. The close association between CD147 and MCTs is consistent with the high level of CD147 expression in metabolically active cells, such as tumor cells. 

Another partner of CD147 involved in the regulation of metabolism is the transmembrane protein GLUT1 that participates in the internalization of glucose. Several reports have demonstrated the interaction of CD147 with GLUT1, and in addition, it has been reported that increased expressions of GLUT1 and CD147 facilitate the entry of glucose into cancer cells to promote tumor glycolysis. GLUT1 overexpression has been correlated with an adverse prognosis in various cancer types as it leads to progression, invasion, and metastasis [52,53]. Overexpression of GLUT1 has been frequently associated with the upregulation of CD147 in several types of tumors, such as papillary renal cell carcinoma (pRCC) [52] or cervical squamous cell carcinoma [54], or melanoma, where it affects aerobic glycolysis favoring metabolic reprograming and tumor growth [55]. In hematologic malignancies, GLUT1 disruption through genetic and pharmacological treatments affects leukemic stem cell (LSC) activity by disrupting energy metabolism, accompanied by increased apoptosis, differentiation, and autophagic activity. In human AML cell lines and in most samples tested from patients with AML, the inhibition of GLUT1 and oxidative phosphorylation (OXPHOS) act synergistically, impairing leukemia cell survival [56]. Although GLUT1 expression has been frequently observed in a wide range of different expressions among the various Hodgkin lymphoma subtypes [57,58], it was not associated with clinical outcomes [57]. In multiple myeloma, an interesting study observed that some myeloma cell lines strongly express GLUT1, but found that the glucose uptake of myeloma cells is largely dependent on GLUT4 [59]. Since the suppression of GLUT1, through RNA interference, induces the inhibition of cell growth and/or death, the possibility arises that different subtypes of GLUT are responsible for glucose uptake in different individual myeloma cells [60]. 

Interactions with integrins are particularly important when considering CD147 signaling. α 3 β 1 and α 6 β 1 integrins were colocalized with CD147 on the cell surface [61]. At least four molecules can mediate interactions between CD147 and integrins: CD98, CD43, MCT4, and galectin-3. The CD98 heavy chain (CD98hc) can form covalent heterodimers with large, neutral amino acid transporter 1 (LAT1) and Asc-type amino acid transporter 2 (ASCT2) that mediate amino acid transport across the plasma membrane. The CD147/CD98 cell surface complex plays a role in energy metabolism, probably by coordinating lactate and amino acid transport. CD147 binds directly to CD98, which in turn binds covalently to amino acid transporters and integrins [62]. CD98 is overexpressed in several hematological malignancies; in AML, CD98 promotes the propagation and lethality of AML by promoting the interaction of leukemic cells with their microenvironment and maintaining LSCs [63] and contributing to the growth of AML and the survival of LSCs [63]. In AML, CD98 is highly expressed on proliferating cells and functions as a chaperone for I-type amino acid transporters, such as LAT1 and LAT2. Furthermore, RNAi depletion of CD147 or CD98hc reduced the cell surface expression of both molecules and reduced cell proliferation [64]. The aberrant regulation of CD98 expression has also been observed in myeloma [65] and in lymphoma [66]. CD98 could also play a role in MDS, since it is known that the bone marrow microenvironment is altered in these pathologies and microenvironmental support is particularly important for the survival of MDS cells, so it is likely that leukemia–niche interactions that are important for the growth of AML play a role in the progression of MDS. 

Among the molecules that bind to CD147 extracellularly and transmit their signals, cyclophilin A has been examined in detail [67]. Cyclophilin A is present intracellularly and is also secreted in response to immunological stimuli. Secreted cyclophilin A exhibits chemotactic activity for neutrophils, eosinophils, and T cells. CD147 has been found to be the major signaling receptor for cyclophilin A and its related molecule, cyclophilin B. Cyclophilin A typically binds to heparan sulfate, a heparin-like polysaccharide, and therefore to CD147 [67,68]. B-cell malignancies, such as multiple myeloma, frequently colonize bone marrow. This colonization is mediated by cyclophilin A and CD147. Thus, cyclophilin A secreted by endothelial cells in bone marrow blood vessels attracts myeloma cells, which strongly express BSG, the major receptor for cyclophilin A.

The interaction between CD147 and P-glycoprotein may be important for chemoresistance [69]. Other proteins also use CD147 as a receptor, such as cyclophilins; platelet glycoprotein VI (GPVI), which intervenes in platelet adhesion; and S100A9, a component of the heterodimeric protein calprotectin, which is released during tissue damage and is implicated in inflammation and metastasis [67,68,69,70]. 

## 3. Regulation of Energy Metabolism in Hematological Malignancies

A distinctive feature of cancer cells is their ability to sustain increased energy needs by adopting metabolic reprograming that allows for a high rate of growth and proliferation [10]. Metabolic dysregulation that is associated to hematologic malignancies, such as leukemias, lymphomas, myelomas, and MDS, involves various mechanisms that activate signaling pathways and regulate the expression of metabolism-related genes. Taking into account the heterogeneity between these HMs, identifying their metabolic signatures and key pathways as potential therapeutic targets may offer specific therapeutic opportunities for the treatment of specific subgroups of HMs [10,71,72,73,74]

Metabolic rewiring is characterized by a faster production of ATP and metabolic intermediates for the biosynthesis of proteins, lipids, nucleotides, and NADPH useful for neoplastic growth, accompanied by the activation of different antioxidant systems to maintain redox homeostasis [10]. Tumor metabolism is characterized by enhanced aerobic glycolysis (“Warburg effect”) for the rapid generation of ATP, instead of producing ATP through oxidative phosphorylation (OXPHOS), and by an increased production of lactic acid, to meet the pressing demands of intensively proliferating tumor cells [75,76]. The increase in glycolysis in cancer cells is not due to a defective mitochondrial respiratory chain, as tumor cells have functional mitochondria that play an important role in energy production. This characteristic metabolic bipotentiality allows cancer cells to change their metabolism from a glycolytic phenotype to an oxidative phenotype and vice versa, allowing the rapid adaptation to the change in the microenvironment and to acidosis and the decreased availability of nutrients and oxygen, which occur during neoplastic development [77]. These metabolic adaptations likely differ between AML subtypes [78] and even within the same patient’s own leukemia cells [79].

CD147 promotes metabolic rewiring by favoring the adaptation of tumor cells, which leads to an advantage of selective growth to some cell subclones and allows cell proliferation. The metabolic regulation of CD147 occurs through its direct or indirect interaction with specific protein partners related to glycolysis, OXPHOS, hypoxia, and lipolysis, which are dysregulated metabolic pathways in cancer, including hematologic malignancies [80,81]. 

### 3.1. Acute Myeloid Leukemia

Acute myeloid leukemia (AML) is a clonal hematopoietic disorder affecting hematopoietic stem and progenitor cells, which results in the blockage of myeloid differentiation and the suppression of hematopoietic functions. AML is the most commonly occurring acute leukemia in adults and its incidence increases with age. As a result of genetical mutations in hematopoietic stem/progenitor cells, AML is a highly heterogeneous disease [82]. An increase in glycolysis has been observed in AML cell lines and in primary human AML blasts. Leukemia cells to ensure high glycolytic levels use amino acids as fuel, in particular, glutamine, glutamate, and proline. Amino acids are converted into metabolic intermediates to support the anaplerosis of the TCA cycle. In addition to amino acid metabolism, AML cells can compensate metabolically by increasing fatty acid metabolism to promote their survival and quiescence [83].

In acute myeloid leukemia (AML), CD147 has already been reported to be overexpressed in AML cells [15]. The knockdown of CD147 in AML cells suppressed growth and proliferation, suggesting the role of CD147 in signaling pathways that contribute to disease pathophysiology [43]. The aberrant expression of CD147 directly affects various tumor metabolic activities, including glycolysis, due to its participation in lactate transport with MCT1 and MCT4 transporters. Through the inhibition of the two MCT transporters, AR-C and SYRO, glycolysis was reduced, resulting in the accumulation of intracellular lactate, ATP depletion, decreased cell pH, and arrest of cell proliferation in AML cells. Although AR-C and SYRO have a different impact on the rate of lactate import and export in leukemia cells, SYRO appears to be more efficient in blocking the lactate flow capacity of leukemia cells by predominantly inhibiting the export of lactate via MCT4 targeting. The CD147-MCT1/4 axis was therefore crucial to support lactate and H+ trafficking and pH balance in AML. In another study in vitro, the antileukemic effects of seven metabolic inhibitors on patient-derived AML cells were compared, targeting various metabolic pathways, including glycolysis (among these, the selective MCT1 inhibitor AZD3965), the pentose phosphate pathway, glutaminolysis, and fatty acid oxidation. The metabolic inhibitors that had the greatest antiproliferative and proapoptotic effects on leukemia cells were found to be the glycolysis inhibitors that exerted a stronger effect on most patients [84].

Song et al. found that AML cell chemoresistance was associated with increased glycolytic activity and low oxidative phosphorylation efficiency. Through expression analyses of HL-60 Adriamycin-resistant cells and HL-60-sensitive cells, the authors showed that the increase in chemoresistance was linked to the aberrant expression of glycolysis-related molecules, such as hypoxia-inducible factor (HIF)-1α, glucose transporter (GLUT)1, hexokinase-II (HK-II), aldehyde dehydrogenase (ALDH), and CD147, in chemo-resistant cells. In addition, the authors demonstrated that the inhibition of glycolysis with 2-DG or 3 BrPA can reverse the chemoresistance of the Adriamycin-resistant HL-60 AML cell line [85,86].

Several studies have reported the physical association of the CD98 protein with CD147 to form a complex on the cell surface of different solid tumors and leukemias, highlighting the role of the CD147/CD98 complex and its contribution to both adhesive signaling and the modulation of metabolic function [62,87]. From the study of this complex in different tumor cell lines, it has been shown that the loss of CD98hc inhibits the absorption of amino acids and glucose and suppresses glycolysis, interfering with the pentose phosphate (PPP) pathway, oxidative stress, and cell cycle arrest [63,88]. CD98 is overexpressed in AML and associated with poor prognosis, demonstrating its crucial role in driving malignant transformation and tumor progression, as was also shown for CD147 in AML [15,20,63]. In addition, CD147 is involved in the engraftment and propagation of primary leukemia cells in their microenvironment favoring tumor progression [63,87]. To further elucidate the role of CD98 in leukemogenesis, Bajaj and colleagues removed CD98 from mouse models of AML, increasing the survival of AML mice, while the proliferative and self-renewal abilities of normal hematopoietic stem cells (HSCs) were reduced in non-tumor mice. In addition, the CD98 knockdown resulted in a decrease in the frequency of leukemic stem cells (LSCs). Furthermore, the authors found that CD98-integrin-mediated signaling and CD98-mediated amino acid transport contribute to AML growth and LSC survival [63].

### 3.2. Chronic Myeloid Leukemia

Biochemical analysis of a panel of imatinib-resistant chronic myeloid leukemia (CML) cell lines and mononuclear cells isolated from CML patients who had relapsed after imatinib treatment showed increased glucose uptake and increased glycolysis. Metformin treatment of these cells exerted antiproliferative effects by inhibiting the CD147 co-binding proteins, MCT1 and MCT4 transporters, leading to the inhibition of lactate export. Metformin-mediated inhibition occurs through the regulation of mTORC1 and HIF-1α. In addition, glucose uptake and ATP production were also inhibited following metformin treatment due to the AMPK-dependent inhibition of GLUT1 and HK-II [89].

Although a direct relationship between CD147 and metabolic regulation in CML are still to be explored, a recent study “in vivo” investigated the antitumor efficacy of a novel CD147-targeting antibody (h4 ^#^ 147D) in three xenograft mouse models that housed chemo-resistant human cell lines from different cancers. In a mouse xenograft model inoculated with KU812 CML cells (with low sensitivity to imatinib), treatment with the anti-CD147 antibody exerted a 99% antitumor effect compared to the imatinib group (TGI 48%), resulting in complete tumor regression. In addition, the analysis of xenograft tumor tissues after treatment with the CD147 antibody highlighted reductions in the levels of CD147 and its binding proteins (CD44, integrin α 3, integrin α 6, and MCT1) [90].

### 3.3. Lymphoma

Lymphomas are part of a complex and highly heterogeneous group of lymphoid tissue neoplasms, which are divided into two main categories, Hodgkin lymphoma (HL) [91] and non-Hodgkin lymphoma (NHL), based on the presence of Reed–Sternberg cells, which is the hallmark of NHL [92]. Non-Hodgkin lymphoma is more common than Hodgkin lymphoma and includes diffuse large B-cell lymphoma (DLBCL), follicular lymphoma (FL), mantle cell lymphoma, and Burkitt lymphoma [92,93].

Several studies have shown that CD147 is overexpressed in different lymphoma and leukemia cell lines [44,94,95].

The involvement of CD147 in the regulation of glycolysis has been demonstrated in anaplastic large cell lymphoma (ALCL), a T-cell lymphoma [55]. A recent study showed that CD147 was differentially expressed in systemic ALK (anaplastic lymphoma kinase)-positive ALCL (ALK+ ALCL) versus ALK-negative ALCL (ALK− ALCL) and normal T cells, as CD147 is a direct target of miR-146. Interestingly, the authors observed that CD147 contributes to the survival and proliferation of ALK+ ALCL cells in vitro and to engraftment and tumor growth in vivo in an ALK+ ALCL-xenotransplant mouse model. The direct involvement of CD147 with glycolysis was demonstrated through the CD147 knockdown in ALK+ ALCL cells, which led to the loss of MCT1 expression, reduction in glucose consumption, and retardation of tumor growth [96]. In another study, the role of CD147 in the progression of T lymphoma was determined by CD147-specific siRNA in the Jurkat T cell line expressing CD147. Silencing CD147 expression causes the decreased proliferation and migration of Jurkat cells and reduces Jurkat cell adhesion to the extracellular matrix fibronectin in vitro [97].

The possibility of CD147’s interaction with other protein partners has favored the exploration of possible therapeutic targets. The interaction of CD147 and CypA is involved in cutaneous T-cell lymphoma (CTCL), a rare T-cell lymphoma. CD147 and CypA are both overexpressed in CTCL cells, blocking these proteins with an anti-CD147 antibody and/or anti-CypA antibody, suppressing the proliferation of CTCL cell lines, both in vitro and in vivo, via the downregulation of phosphorylated extracellular-regulated kinase 1/2 and Akt [98].

### 3.4. Acute Lymphoblastic Leukemia and Chronic Lymphocytic Leukemia

Acute lymphoblastic leukemia (ALL) originates from the transformation and uncontrolled proliferation of B-cell precursors (B-ALL) or T lymphoid progenitors (T-ALL) [99]. ALL is the most prevalent, acute, common childhood malignancy, manifested by an expansion of immature B or T cells. The glycolytic pathway is a critical step in the malignant transformation of B cells, and B-lymphoid repression of glucose uptake and energy supply represents a previously unrecognized metabolic barrier against malignant pre-B-cell transformation, while B-lymphoid transcription factors function as metabolic gatekeepers by limiting the amount of cellular ATP to levels that are insufficient for malignant transformations [100]. CD147 has been previously reported to be overexpressed in several ALL cell lines and in primary cells derived from patients with relapsed acute lymphoblastic leukemia [32]. In this context, the function of CD147 may be to support the glycolytic phenotype typically associated with this disease.

The alteration in glycolysis characterizes chronic lymphocytic leukemia (CLL)-malignant cells [101]. CLL-B-cell interactions with the supportive tissue microenvironment play a critical role in disease pathogenesis. CD147 has been described in CLL-B cells, and its expression has been functionally linked to leukemic clone activation during disease progression [102], as well as with CLL-B cell migration and BM colonization [17]. The expression of CD147 is heterogeneous and was detected along with increased MMP9/MMP2 activity in plasma samples from CLL patients compared to healthy subjects. In this context, it was shown that CD147 is associated with MMPs production by tumor and stromal cells, highlighting the role of CD147 and MMPs in the pathophysiology of CLL [103].

### 3.5. Multiple Myeloma and Myelodysplastic Syndrome

Multiple myeloma (MM) is a cancer that affects a particular type of cell: plasma cells in the bone marrow. The abnormal and uncontrolled multiplication of a plasma cell gives rise to a population (a clone) of cancer cells [104,105]. Myelodysplastic syndrome/myeloproliferative neoplasms (MDS/MPNs) are a group of chronic onco-hematological diseases characterized by the coexistence of alterations in cell maturation, typically seen in MDS, with an increase in cell proliferation in the bone marrow and peripheral blood, also seen in chronic MPNs. MDS and MPNs are most common in elderly patients [106,107]. Myeloma metabolism is characterized by increased glycolysis and the use of fatty acid oxidation that promote tumor survival and proliferation. CD147 is involved in the modulation of the aerobic glycolysis of multiple myeloma (MM) through the regulation of MCT1 expression and lactate exportation [108]. Studies on MM have shown that CD147 expression is increased, and this increase accompanies disease progression [109]. The transporters MCT1 and MCT4, partners of CD147 in lactate transport, were also overexpressed in MM, but only MCT1 appears to play a role in myeloma cell proliferation and to promote tumor growth in an acidic microenvironment [108]. In addition, single nucleotide polymorphisms (SNPs) in the gene encoding for CD147 (*BSG* SNPs) and MCT1 (*SLC16A1* SNPs) have been shown to affect MM survival [110]. The development and progression of myeloma are also linked to its ability to modify the behavior of surrounding cells so that they make the tumor microenvironment more favorable for its growth and survival. From this point of view, MM is a type of cancer that, more than others, needs the support of its own microenvironment, which provides growth signals and survival factors. In addition, interactions with bone marrow stromal cells and alterations in oxygen metabolism contribute to MM cells’ resistance to chemotherapy treatments.

Several studies have highlighted the involvement of CD147 in the response to the chemotherapy treatment of MM [111]. Among these, the study by Eichner et al. found that immunomodulatory drugs (IMiDs), such as thalidomide and its derivatives lenalidomide and pomalidomide, commonly used for the treatment of MM and myelodysplastic syndrome (MDS) with isolated chromosome 5q deletion (del(5q) MDS), which constitutes the only MDS subtype clearly defined by a cytogenetic alteration, bind Cereblon (CRBN), a substrate receptor of the ubiquitin ligase complex CRL4, mediating antitumor and teratogenic effects [19]. From the analysis of primary MM and CD34^+^ MDS cells treated with IMiD, it was found that CRBN functions via a ubiquitin-independent chaperone-like mechanism to mediate the folding and maturation of CD147 and MCT1 proteins, thereby allowing the activation of the CD147–MCT1 transmembrane complex. IMiD treatment abrogates this mechanism in a competitive manner to mediate their antitumor and teratogenic activities. As a result, IMiD-sensitive MM cells lose CD147 and MCT1 expression after being exposed to IMiDs, while IMiD-resistant cells retain their expression. In addition, del(5q) MDS cells have an elevated CD147 expression, which attenuates after IMiD treatment. This study showed that the elimination of CD147 from zebrafish restores the teratogenic effects of thalidomide exposure [19].

## 4. CD147 in a Leukemic Microenvironment

The impact of metabolism on the leukemic microenvironment is crucial for the proliferation of hematopoietic neoplasms. The regulation of the leukemic microenvironment is complex; the so-called “niches” in the bone marrow (BM), lymph nodes, and secondary lymphoid organs are composed of different non-tumor cell types, which can interact with each other and with tumor cells, providing regulatory factors and growth factors to proximal malignant cells supporting their growth [112,113,114]. Several studies have also revealed that the microenvironment is characterized by a hypoxic environment, and that the intercellular interaction of leukemia cells with stromal cells can influence cellular metabolism and consequently play an important role in the growth and survival of leukemia cells [115,116,117,118]. In this context, CD147 is at the center of a crossroads, directly or through signaling to downstream molecules, and favors the hypoxic and metabolic adaptation of leukemia stem cells [114].

The analysis of the immunophenotypic expression of mesenchymal stromal cells derived from bone marrow and umbilical cord blood showed that the CD147 protein is expressed by tissue-specific stem cells [119]. In addition, CD147 has been previously shown to regulate the expression of hypoxia-inducible factor 1 α (HIF1-α) and HIF-2α, factors involved in angiogenesis and the hypoxic and metabolic reprograming of different types of neoplasms [120,121,122].

Interestingly, Bougatef et al. observed that HUVEC endothelial cells express endogenous CD147, and showed that CD147 expression is responsible, through the transcription factor HIF-2α, for the increased production of soluble VEGF isoforms and its main receptor VEGFR-2, both in vitro and in experimental tumor models in vivo [123]. The authors also observed an increase in VEGFR-2 “in vivo” in a mouse model of xenograft tumors that had an altered level of CD147 expression.

In the BM microenvironment, leukemia cells can be favored by the increase in VEGF, as this factor regulates the expression of the lactate transporter MCT1, allowing the intracellular transport of lactate to be used for the production of energy and nutrients useful for tumor growth [124]. The dependence on this factor also drives the localization of leukemia cells in the marrow according to the VEGF gradient. Through the analysis of BM samples from patients with M0-M5 AML, which express high levels of MCT1, it was observed that leukemia cells prefer to localize near the bones (paratrabecular region) of the BM, where VEGF production is active [124].

Regarding metabolites in the leukemic microenvironment, it is known that lactate, in addition to being present inside cells to be used for energy production, can also be used as a signaling molecule through the lactate receptor G-protein-coupled receptor 81 (GPR81) in physiological conditions [125]. Lactate-GPR81 signaling had previously been observed to drive the growth and repopulation of breast cancer cells [126].

A recent study, through metabolomics analysis, identified metabolites in extracellular fluid derived from BM biopsies of AML patients. The result of these investigations showed that the levels of extracellular metabolites were elevated in the BM of AML patients, including lactate. The increase in lactate is a critical factor in the dysfunction of the BM microenvironment induced by AML and is responsible for the progression of leukemia. The metabolic characteristics make the leukemic bone marrow microenvironment (BMME) unique [127].

The expression of CD147 on leukemia cells’ surface is also important for the preferential homing of the BM. In an interesting study, using multiple myeloma (MM) as a model of a B-cell malignancy that frequently colonizes the BM, it was shown that bone marrow endothelial cells (BMECs) secrete cyclophilin A (eCyPA), which promotes the migration, proliferation, and colonization of leukemic cells through binding to its receptor, CD147, on MM cells’ surface. The study revealed an altered level of eCyPA in BM serum compared to peripheral blood (PB) in people with MM, and blocking eCyPA-CD147 suppresses BM homing and tumor growth in a mouse model of MM xenotransplantation. eCyPA also promoted the migration of chronic lymphocytic leukemia (CLL) and lymphoplasmacytic lymphoma (LPL) cells, two other malignant B-cell neoplasms that colonize BM and express CD147 [17].

It should also be considered that the niche, with its favorable microenvironment, constitutes a refuge for leukemia stem cells (LSCs) responsible for relapse, which, in this environment, are able to survive and proliferate after treatment with chemotherapy [128,129]. Gastel et al. presented an analysis of metabolomics in AML LSC cells, revealing a distinct metabolic profile in the pre-chemo versus post-chemo relapse phase [130].

Another study observed that CML LSCs use the hypoxic niche to escape treatment with imatinib. In detail, the hypoxic environment promoted the upregulation of hypoxia-inducible factor 1 α (HIF1-α) to allow survival independent of BCR-ABL. Along with HIF1-α, other pro-survival genes are also stimulated that are preferentially upregulated by hypoxia in CML compared to normal CD34+ blood progenitors [131].

## 5. Therapeutic Targeting of CD147

Numerous studies have ascertained the crucial role played by the aberrant expression of CD147 in cancer progression, demonstrating the potential application of CD147 not only as a potential diagnostic marker of the disease [39], but also as an effective therapeutic target for the development of new anticancer therapies (Table 2).

The basic approach for targeted therapy involves the downregulation of CD147 protein expression, with the aim of blocking its function by promoting leukemic cell death. Several preclinic studies have reported a decreased expression of the CD147 protein through treatment with small compounds of synthetic or natural origin, and with RNAi [15,43,132]. In addition, new anti-CD147 antibodies [90,133,134] or antigenic peptide vaccines [90,133,135] have been designed in order to block CD147 function. We have reported previously a study on the small molecule AC-73 able to block the dimerization of CD147, which led to the inhibition of the proliferation of leukemia cells and the induction of autophagy. In addition to an antiproliferative action, AC-73 was able to improve sensitivity to chemotherapy treatment and allowed a decrease in the dosage of chemotherapy (arabinosylcytosine and arsenic trioxide) [15]. Given the widespread expression of CD147 in normal hematopoietic cells and its function as a T-lymphocyte activating antigen [136], the use of this inhibitor should be carefully evaluated. Indeed, it has recently been found that the inhibitor AC-73 suppresses the activation and recruitment of T cells in CVB3-induced acute viral myocarditis [137].

Recently, the study by Zou et al. showed that pseudolaric acid B (PAB), a bioactive compound of plant origin, was able to inhibit CD147 in several AML leukemia lines, confirming the role of CD147 in AML proliferation [43]. However, for the clinical application of these new compounds, further characterization will be necessary to evaluate the therapeutic potential and optimal dosing strategies.

The therapeutic intervention based on blocking the CD147 function with mAb could be promising, but although significant clinical progress has been made in the development of CD147-directed monoclonal antibodies for the treatment of different types of neoplasms, including hepatocellular carcinoma (HCC) [138,139], cervical cancer [140], lung cancer [141], and other solid tumors [16], few studies have explored the application of anti-CD147 antibodies in the field of hematological malignancies. Fukuchi and colleagues generated a humanized anti-CD147 antibody, which showed a potent antitumor effect in three mouse xenograft models, including the imatinib-insensitive CML model. Particularly, the authors observed complete tumor regression in the group of xenograft mice treated with the antibody compared to the group of mice treated with the drug imatinib [90]. In an interesting study, K562/ADR cells resistant to the common chemotherapy Adriamycin were found to have significantly higher levels of P-glycoprotein (P-gp) and CD147 than drug-free K562/ADR cells. Using monoclonal antibodies against CD147, mRNA and protein levels of P-gp and MDR1 were significantly reduced. The activation of CD147 by the anti-CD147 antibody also reduced CD147 expression [134].

Recently, immunotherapy has made important contributions to cancer treatments. Chimeric antigen receptor (CAR) T-cell therapy currently represents an important therapeutic success for the treatment of relapsed and/or refractory B-ALL and relapsed specific large B-cell lymphoma [142]. CART cells are engineered to specifically recognize target tumor antigens, such as CD19, CD20, and CD22, in B ALL, and to rapidly activated to kill tumor cells. Despite the notable successes achieved with CART-based therapies in the treatment of acute lymphoblastic leukemia (B ALL), they have not been recognized as effective for the treatment of acute lymphoblastic leukemia (T-ALL); therefore, it is necessary to explore new antigenic targets for leukemia acute lymphoblastic disease (T ALL). As CD147 had elevated expression levels in tumor T cells from patients with ALL T-cell and T-cell lymphoma, it was considered a potential antigenic target of T-CAR cell therapy for T-ALL. CD147-CAR T cells, which contain a humanized single-chain variable fragment targeting CD147, showed potent antitumor activity against the human T-ALL cell line and T-ALL blasts, releasing high levels of cytokines in the process. However, CD147-CAR T cells have shown a potential safety concern against normal human cells and CD147-deficient cells, which must be considered for future therapy [143]. These results highlight the therapeutic potential of CD147-CAR-modified immune cells for patients with ALL.

Chemotherapy drug resistance represents a major problem in cancer therapy that favors treatment failure and patient relapse. An important challenge for clinical research is to identify patients who will develop resistance to therapy and to devise more effective alternative treatment strategies. Several studies suggested that CD147 glycoprotein plays a crucial role in resistance to anticancer drugs. Different tumors, including hematologic malignancies, showned a positive correlation between the level of CD147 expression and resistance to different chemotherapy drugs. In lymphoid neoplasms, CD147 has been observed to increase resistance to chemotherapy drugs through the formation of a complex consisting of CD147, the endothelial lymphatic vessel hyaluronic acid receptor-1 (LYVE-1), and a drug transporter known as breast cancer-resistance protein/ABCG2 (BCRP), located on the cell surface of a primary effusive lymphoma tumor cell line (PEL). Targeting every single component of the CD147/LYVE-1/BCRP complex with RNAi showed the interdependent roles of the three molecules in chemotherapy resistance [144]. These findings highlight that CD147-targeted therapy could be a potential approach to bypass drug resistance.

**Table 2 ijms-25-09178-t002:** CD147-based therapies for hematological malignancies.

Name	Mechanism of Action	Cell Lines/Tumor Xenograft	Effects	References
**siRNA**	Silencing	AML cell line	Inhibits leukemic cell proliferation and increases chemosensitivity to Adriamycin	[132]
**siRNA**	Silencing	Lymphoma cell line	Increases chemosensitivity	[144]
**AC-73**	Disrupts CD147 dimerization	Cell lines and primary AML blasts	Inhibits leukemic cell proliferation and increases chemosensitivity to ARA-C and ATO	[15]
**PAB**	CD147 inhibitor	Cell lines and primary AML blasts	AML cell apoptosis	[43]
**h4^#^147D**	Humanized anti-CD147 antibody	Xenograft CML mouse models	Strong antitumor effects with tumour regression	[90]
**MEM-M6/6**	Monoclonal CD147 antibodies (mAbs)	CML cell line	Decreased expression of CD147 and causes downregulation of P-gp	[134]
**CD147-CAR T cells**	(CAR) T-cell therapy	T-ALL cell line and T-ALL blasts	Antitumor activity	[143]

## 6. Conclusions and Future Directions

CD147 performs multiple metabolic functions that profoundly influence cellular metabolism and the tumor microenvironment in hematologic malignancies. The high expression of this glycoprotein in hematological malignancies represents metabolic vulnerability that can be exploited therapeutically. Its interaction with monocarboxylate transporters or other partners, such as CD98 or Glut1, and the regulation of energy metabolism are crucial for the proliferation and survival of leukemia cells. Understanding these mechanisms offers new opportunities for the development of CD147-targeted therapies, potentially improving the treatment of hematologic malignancies. However, while targeting CD147 may be considered a future therapeutic strategy, the potential risks for clinical application should be taken into consideration, given the widespread expression of CD147 in normal hematopoietic and non-hematopoietic cells and its function as a lymphocyte-activating antigen that potentially negatively affects immune cells [136]. Anti-CD147 monoclonal antibody-targeted therapy for HM is a promising strategy. The cost of use and security represent a major challenge. Furthermore, the widespread expression of CD147 requires toxicity studies for mAbs in a cross-reactive species. The inhibition of T-cell activation and/or depletion should be investigated for new CD147 mabs used to treat HM. Overcoming these problems will make CD147 prominent in the treatment of HM. 

Other challenges in the treatment of hematological malignancies concern the critical issues related to CAR-T therapies, which include on-target off-tumor effects, the immunosuppressive microenvironment, and toxicities associated with CAR-T cells, such as strong cytokine-induced effects and neurotoxicity.

However, CD147 undoubtedly has a strong impact on tumor cell invasion, proliferation, angiogenesis, and metabolism by mediating pro-survival signals and multidrug resistance, which favor leukemogenesis. Future research should focus on overcoming the critical issues and exploring new therapeutic approaches based on CD147, with the aim of translating this knowledge into clinical benefits for patients

## Figures and Tables

**Figure 1 ijms-25-09178-f001:**
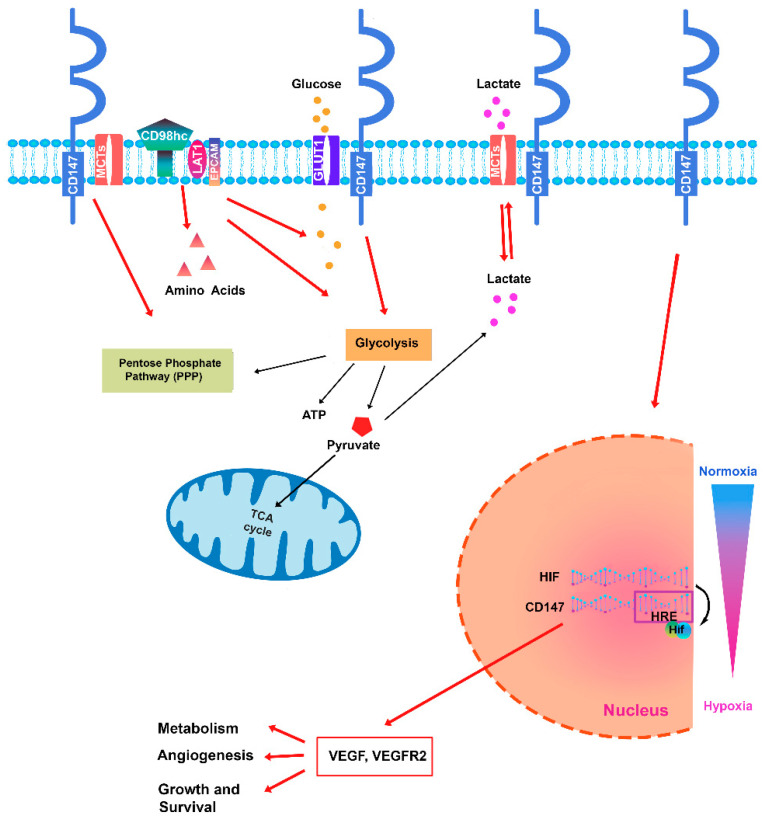
Regulation of glycolytic metabolism by CD147-associated partners: a schematic overview to illustrate the regulation of glycolytic metabolism in hematological malignancies.

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
