# Peer review of "Metabolic Function and Therapeutic Potential of CD147 for Hematological Malignancies: An Overview"

_ijms, 2024, doi:10.3390/ijms25179178_

Round 1

Reviewer 1 Report

Comments and Suggestions for Authors

This is an interesting and well-written review article. It summarizes the role of CD147 for cell metabolism in hematological malignancies. The authors also discuss the possibility that CD147 could be used as targeting structure for tumor therapy. I have a few suggestions to further improve the quality of this article.

1.       It would make more sense to combine part 2 and part 4 (e.g. include 2.1. Acute Myeloid Leukemia as introduction for 4.1. Acute myeloid leukemia).

2.       CD147 has a broad expression on hematopoietic and non-hematopoietic cells. It is not true that CD147 is not expressed on T cells (line 218, see e.g. T cell activation-associated epitopes of CD147 in regulation of the T cell response, and their definition by antibody affinity and antigen density Int Immunol . 1999 May;11(5):777-86. doi: 10.1093/intimm/11.5.777). Therefore, CD147 targeting might also harm immune cells. This has to be mentioned and should be discussed in part 6 (e.g. Inhibitor of CD147 Suppresses T Cell Activation and Recruitment in CVB3-Induced Acute Viral Myocarditis doi: 10.3390/v15051137, CD147 regulates antitumor CD8+ T-cell responses to facilitate tumor-immune escape  Cellular & Molecular Immunology (2021) 18:1995–2009; https://doi.org/10.1038/s41423-020-00570-y).

3.       Please use the correct name “MCT= monocarboxylate transporter” and not “monocarboxylated transporter” (abstract line 20, text line 57…)

4.       “CD147 binds directly to CD98”… (line 267)- please explain (1-2 sentences) the role of CD98.

5.       A table which summarizes the expression/glycosylation/prognoistic value in different hematological diseases would be helpful

6.       “cancer cells have functional mitochondria, although they have reduced activity”.. (line 316). This statement holds not true for all tumor cells. Some tumors even have more mitochondrial activity (e.g. Slow TCA flux and ATP production in primary solid tumours but not metastases Nature volume 614, pages349–357 (2023)).

Author Response

Response to Reviewers

Manuscript ID: ijms-3156075.

Manuscript Title: Metabolic function and Therapeutic Potential of CD147 for Hematological Malignancies: an overview.

Authors: Spinello I, Labbaye C and Saulle E.

We wish to thank the reviewers for the helpful and constructive comments. We believe we have addressed all of the major and minor remarks raised by the two reviewers. Please find the detailed responses below and the corresponding revisions/corrections highlighted/in track changes in the re-submitted files.

Response to Reviewer 1

Comments 1: It would make more sense to combine part 2 and part 4 (e.g. include 2.1. Acute Myeloid Leukemia as introduction for 4.1. Acute myeloid leukemia).

Response 1:   We agree with the reviewer and to give more relevance to the role of CD147 in various pathological conditions, in the revised version of the manuscript we have included each paragraph of section 2 as an introduction of individual hematological diseases in section 4 (section 3 in the revised manuscript).

Comments 2: CD147 has a broad expression on hematopoietic and non-hematopoietic cells. It is not true that CD147 is not expressed on T cells (line 218, see e.g. T cell activation-associated epitopes of CD147 in regulation of the T cell response, and their definition by antibody affinity and antigen density Int Immunol . 1999 May;11(5):777-86. doi: 10.1093/intimm/11.5.777). Therefore, CD147 targeting might also harm immune cells. This has to be mentioned and should be discussed in part 6 (e.g. Inhibitor of CD147 Suppresses T Cell Activation and Recruitment in CVB3-Induced Acute Viral Myocarditis doi: 10.3390/v15051137, CD147 regulates antitumor CD8+ T-cell responses to facilitate tumor-immune escape  Cellular & Molecular Immunology (2021) 18:1995–2009; https://doi.org/10.1038/s41423-020-00570-y).

Response 2:  We agree with the reviewer that CD147 expression in activated T cells was not correctly reported as stated in the cited publication (reference 44, Discussion section).  In the revised version of our manuscript, we have specified "Schmidt's study reports an overexpression of CD147 also in several lymphoma and leukemia cell lines, as well as in normal activated T cells."  Text line 219-220.  We have also mentioned the data of CD147 expression on T cells reported by the cited authors in text line 227-230 “Interestingly, in resting (un-activated) CD3+ T cell samples, the authors did not detect any expression of CD147 at the protein level; this result was confirmed by immunohistochemical analysis of normal T cells in the interfollicular area in normal lymph nodes where CD147 expression was negative in contrast to positive expression in B cells of the germinal center and mantle area ”.  As requested by the reviewer, in section 6 (section 5 in the revised manuscript) line 601-605, we highlighted the critical issues related to the clinical use of CD147 inhibitors “Given the widespread expression of CD147 in normal hematopoietic cells and its function as a T-lymphocyte activating antigen [136], the use of this inhibitor should be carefully evaluated. Indeed, it has recently been found that the inhibitor AC-73 suppresses the activation and recruitment of T cells in CVB3-induced acute viral myocarditis. [137].” 

Comments 3: Please use the correct name “MCT= monocarboxylate transporter” and not “monocarboxylated transporter” (abstract line 20, text line 57…..)

Response 3: As requested by the reviewer, we have used the correct name “monocarboxylate transporter” in abstract line 21, text line 61 and 666 (outlined in green).

Comments 4: “CD147 binds directly to CD98”… (line 267)- please explain (1-2 sentences) the role of CD98.

Response 4: In our revised version of the manuscript, we explain in detail this point (text lines 283 to 287)

“The CD98 heavy chain (CD98hc) can form covalent heterodimers with large neutral amino acid transporter 1 (LAT1) and Asc-type amino acid transporter 2 (ASCT2) that mediate amino acid transport across the plasma membrane. The CD147/CD98 cell surface complex plays a role in energy metabolism, probably by coordinating lactate and amino acid transport”.

Comments 5: A table which summarizes the expression/glycosylation/prognoistic value in different hematological diseases would be helpful

Response 5: As requested by the reviewer 1, we have included in section 2.1 of the revised version of our manuscript, a new table (Table1) summarizing data on CD147 expression in different hematological malignancies.

Comments 6:  “cancer cells have functional mitochondria, although they have reduced activity”.. (line 316). This statement holds not true for all tumor cells. Some tumors even have more mitochondrial activity (e.g. Slow TCA flux and ATP production in primary solid tumours but not metastases Nature volume 614, pages349–357 (2023)).

Response 6:  we have revised the line 335 with” The increased glycolysis in tumor cells is not due to a defective mitochondrial respiratory chain, as tumor cells have functional mitochondria that play an important role in energy production in tumor cells, maintaining reduced activity in some tumors."

Reviewer 2 Report

Comments and Suggestions for Authors

This manuscript offers an insightful discussion on the role of CD147 in metabolic dysfunction and its potential as a therapeutic target in hematologic malignancies. However, before the manuscript can be considered for acceptance, I have several major concerns that need to be addressed:

1.     Formatting Issues: The manuscript contains multiple formatting errors that need careful correction. For example, the sentence on lines 540 and 541 is improperly split between two columns. Such issues detract from the manuscript's professionalism and should be thoroughly reviewed and corrected throughout.

2.     Section 2 Revisions: The content in Section 2 regarding hematologic malignancies should be revised to establish a stronger connection to CD147. Currently, the section reads more as a general overview of hematologic malignancies without adequately linking these discussions to the role of CD147. The authors should ensure that the section is more focused on how CD147 is implicated in these conditions, thereby making the content more relevant and cohesive.

3.     Section 6 Improvements: In Section 6, while the authors list several therapeutic opportunities targeting CD147, the section lacks a clear rationale for these strategies. The absence of a detailed explanation of the design principles and the strategic approaches undermines the readability and impact of this part. The authors should elaborate on the rationale behind targeting CD147 and discuss the strategic considerations involved in developing these therapeutic approaches.

4.     Discussion on Challenges and Outlook: The manuscript would benefit from a more in-depth discussion of the challenges and future prospects of targeting CD147 as a therapeutic approach. The current discussion at the end of the manuscript is too brief and does not provide sufficient critical analysis. A more comprehensive examination of the potential obstacles and the outlook for CD147-targeted therapies would greatly enhance the manuscript’s contribution to the field.

Comments on the Quality of English Language

The language of the manuscript is fine. No further revise is required.

Author Response

Response to Reviewers

Manuscript ID: ijms-3156075.

Manuscript Title: Metabolic function and Therapeutic Potential of CD147 for Hematological Malignancies: an overview.

Authors: Spinello I, Labbaye C and Saulle E.

We wish to thank the reviewers for the helpful and constructive comments. We believe we have addressed all of the major and minor remarks raised by the two reviewers. Please find the detailed responses below and the corresponding revisions/corrections highlighted/in track changes in the re-submitted files.

Response to Reviewer 2

Comments 1: Formatting Issues: The manuscript contains multiple formatting errors that need

careful correction. For example, the sentence on lines 540 and 541 is improperly split between two columns. Such issues detract from the manuscript's professionalism and should be thoroughly reviewed and corrected throughout.

Response 1: As suggested by the reviewer, we revised the formatting issues in lines 540 and 541.

Comments 2: Section 2 Revisions: The content in Section 2 regarding hematologic malignancies

should be revised to establish a stronger connection to CD147. Currently, the section reads more as a general overview of hematologic malignancies without adequately linking these discussions to the role of CD147. The authors should ensure that the section is more focused on how CD147 is implicated in these conditions, thereby making the content more relevant and cohesive.

Response 2: To give more cohesion and relevance to the role of CD147 in various pathological conditions, in the revised version of the manuscript we have used/included the individual paragraphs of section 2 as an introduction in section 4 (section 3 of the revised manuscript).

Comments 3:  Section 6 Improvements: In Section 6, while the authors list several therapeutic

opportunities targeting CD147, the section lacks a clear rationale for these strategies. The absence of a detailed explanation of the design principles and the strategic approaches undermines the readability and impact of this part. The authors should elaborate on the rationale behind targeting CD147 and discuss the strategic considerations involved in developing these therapeutic approaches.

Response 3: As requested by reviewer 2, we have attempted to clarify the rationale behind the adoption of these therapeutic strategies and to explain the mechanisms underlying these strategies in more detail at the various points where therapeutic strategies based on targeting CD147 have been discussed in Section 6 (Section 5 in the revised manuscript).

Comments 4: Discussion on Challenges and Outlook: The manuscript would benefit from a more in-depth discussion of the challenges and future prospects of targeting CD147 as a therapeutic approach. The current discussion at the end of the manuscript is too brief and does not provide sufficient critical analysis. A more comprehensive examination of the potential obstacles and the outlook for CD147-targeted therapies would greatly enhance the manuscript’s contribution to the field.

Response 4: As requested by reviewer 2, we have expanded the discussion in section 7 (Section 6 in the revised manuscript), on the strategic opportunities and possible critical issues of CD147-based therapies in the application on hematological malignancies.

Round 2

Reviewer 2 Report

Comments and Suggestions for Authors

The manuscript can be accepted.